# Utilization of Job Demands-Resources (JD-R) Theory to Evaluate Workplace Stress Experienced by Health Care Assistants in a UK In-Patient Dementia Unit after 10 Years of National Financial Austerity (2008–2018)

**DOI:** 10.3390/ijerph20010065

**Published:** 2022-12-21

**Authors:** Christopher Chigozie Udushirinwa, Andrew McVicar, Julie Teatheredge

**Affiliations:** Department of Health, Education, Medicine and Social Care, School of Nursing, Chelmsford Campus, Anglia Ruskin University, Chelmsford CM1 1SQ, UK

**Keywords:** workplace stress, job stress, health care assistants, healthcare support worker, HCA, social therapist, dementia

## Abstract

Aims: Workplace stress for support workers in UK hospitals (Health Care Assistants; HCAs) is poorly understood. This study explores experiences of HCAs working in a National Health Service in-patient dementia unit after 10 years of national financial austerity (2008–2018). Design: Qualitative evaluation. Methods: 15 HCAs (42%) from a specialist dementia care Unit were interviewed. Interviews were guided by UK Health & Safety Executive published dimensions of work stress. Framework analysis was applied to interview transcriptions, corroborated by a follow-up focus group (6 HCAs). Post hoc interviews with 10 nurses were later introduced to obtain a balanced view of teamwork on the Unit. Results: Health care assistants were altruistic regarding demands of dementia care but otherwise negative of most aspects of their work environment. Staff shortages had increased job demands: workload, poor shift rotas, and excessive reliance on inexperienced agency staff. According to HCAs, job resources of the care team were in significant deficit: nurses in charge were perceived as poor team leaders, had poor interpersonal skills, lacked respect for experienced HCAs, and deemed to be frequently absent from the ward so failing to support carers. HCAs’ lack of decision-latitude exacerbated the situation. In contrast, nurses interviewed did not recognise the teamwork issues raised by HCAs, who were considered obstructive, unsupportive, lacked awareness of nurses’ responsibilities, and of insights how understaffing meant excessive administration and time required to support patients’ relatives. Such dissonant inter-group views caused considerable friction and exacerbated the work pressure. Conclusion: Study outcomes spotlighted impacts of socioeconomic issues for HCAs. Staff shortage, exacerbated by financial austerity measures (pre-COVID pandemic), increased job demands for HCAs but their psychosocial job resources were in serious deficit, so putting them at risk of burnout. Inter-group relations are key for a collaborative ethos, and are amenable to interventions. Such difficulties should not be allowed to fester.

## 1. Introduction

Around the time of this present study approximately 40% of all staff sickness absence in the UK National Health Service (NHS) was related to work stress [1]. The risk of stress-related ill-health is very high for carers of people with dementia [2,3]. Dementia is a neurodegenerative disease characterised by loss of memory, depression, impaired communication, confusion, poor judgement, disorientation and difficulty in swallowing, walking and speaking [3,4,5]. Consequently, most patients require a high degree of assistance with their activities of daily living [4,6]. Adverse patient behaviours may lead them to be resistive [7] even in some to the extent of becoming abusive or aggressive [8]. Incessant psychosocial and physical engagements whilst caring for in-patients with dementia therefore are distinctive sources of stress for carers [9].

In addition, existing difficulties in recruitment and retention of staff in recent years have led to significant staff shortages that were exacerbated by financial austerity measures introduced in the UK in 2008 [10]. In mental health care austerity led to an estimated 12% decrease in qualified, mental health staff posts by the time measures were relaxed in 2019 [11]. One consequence for the NHS was increased reliance on support workers who are less costly than nurses but they do not have decision-latitude in practice [10,11].

Dementia care in the United Kingdom is delivered by a team comprised primarily of unqualified support workers, mainly health care assistants (HCAs), working alongside nurses. Doctors are not regularly based on the wards, only visiting patients for review and/or to admit them. HCAs work directly with patients but rely on the nurse in charge of the team to ensure a fair distribution of work and to provide a strong network of support and risk management. While evaluations have been made of the NHS work environment for qualified staff [12] the situation for support workers is poorly understood [13]. The comprehensive study by Schneider et al. [14] of HCAs working in specialist dementia settings identified concerns over staffing levels and marginalisation of HCAs by qualified colleagues, but good team relationships ensured effective care delivery and supported the well-being of team staff.

The recruitment of care staff to the National Health Service (NHS) had been difficult for many years prior to the time that the study of Schneider et al. (2010) was undertaken [15]. Introduction in 2008 of financial austerity measures by the UK Government included reduced funding to the NHS and led to significant cuts in care staff, so exacerbating the shortage [10]. When the austerity period was declared ‘over’ in 2019, figures identified 35,000+ advertised vacancies of care staff in nursing and midwifery alone [11] and in mental health care there was an estimated 12% decrease in qualified, clinical staff posts between 2009 and 2020 [11].

One response to the significant shortage of nurses was an increased reliance on support workers, who are less costly to the NHS [10]. Recruitment initiatives introduced in 2014, part way through the austerity period, increased the numbers of support workers in the NHS of around 11% by its end in 2019 [10]. Overall, between 2009 and 2021, which incorporated the austerity period, there was a 31% increase in support staff [11]. However, those figures included the recently introduced Assistant Practitioners whose status and roles are situated between qualified nurses and unqualified HCAs [16]. The increase in HCAs over the period was actually just 2%, which was modest in comparison to the underlying chronic shortage of staff. Further, recruitment also varied according to clinical specialism. In mental health care the numbers of support workers actually decreased by −0.1% [10].

The picture therefore is one of a worsened understaffing of the NHS, including of HCAs, as a consequence of financial austerity measures imposed by the UK Government. Despite this situation little research has been conducted into issues for HCAs. In 2021 a systematic review by the present authors (2010–2020) (*submitted for publication*) identified just six studies that specifically related to HCAs working in specialist dementia care. Five studies focused on specific issues including emotional labour of care [17], resistance of patients to care [7,18], in-group identity as an obstacle to teamwork [19], and demotivating factors linked to intention to leave [20]. Only Schneider et al. [14] had presented a comprehensive evaluation of the work environment for HCAs. As noted above, their findings acknowledged the challenge of understaffing but generally were positive regarding team relationships and the quality of care. In view of the impact of austerity on staffing it appears unlikely those findings represent the more contemporary situation of HCAs.

This paper reports a comprehensive evaluation of the workplace for HCAs working in an NHS dementia care Unit in late 2018 shortly before austerity measures were relaxed. The COVID-19 pandemic began soon after austerity was declared ‘over’ (in 2019) and so this study was timely to provide insight into the situation just before that crisis. Despite subsequent staffing initiatives to address the crisis, the levels of staffing have remained well below targets required to meet current demands [11] and issues reported here are likely to still have relevance.

### Aim

By understanding better the challenges faced by HCAs in dementia care this qualitative study sought to answer the research question

‘What perceptions do Health Care Assistants have of workplace stress in an in-patient dementia Unit?’

The aim was to apply Job Demands-Resources theory [21] to evaluate the work environment for HCAs in an NHS specialist dementia care Unit after almost 10 years of financial austerity in the UK. The JD-R developed from two well-established models of stress in the workplace: the Job Demands-Control-Support model [22], and the Effort-Reward Imbalance model of Siegrist [23]. Job demands relate to those organisational aspects that require physical and psychological effort or skills and, when perceived to be high, are predictors of burnout, low commitment and high turnover [24]. Job resources refer to aspects that help the individual to meet work goals, reduce work demands and/or stimulate personal growth and development. With subsequent incorporation of personal resources into the model [25] it provides a comprehensive, generic model with particular relevance for workplace evaluation and especially stress management in high-demand settings. It provides a ‘balance’ model in that the job resources act to buffer the negative impacts of perceived high job demands [21]. Whereas work environments might be found to vary between organisations, the theory has the flexibility for application to specific professions. For example, in health care an evaluation of the job demands made on care staff would not be accurate without incorporating the emotional demands arising from delivering patient care [26]. The JD-R especially has utility to identify factors that predict job turnover of nurses [27,28].

## 2. Materials and Methods

This study took place in a specialist in-patient dementia care Unit of a large regional hospital. Data were collected during late 2018 after ten years of financial austerity for the UK National Health Service. At the time of the study the Unit comprised 46 beds; all were occupied. Twenty four HCAs were employed in substantive posts. The Unit also employed temporary, agency HCAs some of whom worked regularly and preferentially on the Unit and so were considered as part of the study population.

A qualitative design applied individual interviews with HCAs, followed up by a focus group convened to seek validation of the interview outcomes and to encourage further relevant discussion.

Findings (described below) included what appeared to be significant conflict between experiences of HCAs and nurses working within the care teams. In such a setting, some researchers [29,30] have recommended inclusion of both staff groups when evaluating the work environment. Accordingly, although the focus of this study was on HCAs experiences a post hoc decision was taken to interview a group of nurses from the Unit specifically to provide a more meaningful evaluation of team relationship issues.

### 2.1. Recruitment

An invitation letter and participant information sheet were emailed to the Unit manager who, as gatekeeper, forwarded documentation to all 24 HCAs employed by the Unit and to 12 agency personnel who were very familiar with it. Hardcopies of the documentation were also made available in the Unit staff area.

HCAs interested in taking part were asked to contact author CU directly by email. Completed forms were returned 1–2 weeks later. 15 agreed to participate.

Following the interview phase, HCAs were sent a new participant information sheet via the gatekeeper and invited to take part in a focus group. Eight responded to author CU and all were invited to attend and to complete a consent form, but just 6 eventually took part of whom three participants had previously been interviewed individually, three had not.

Post hoc recruitment of 10 nurses for interview followed the same pattern of recruitment via the gatekeeper. Findings from the interviews with HCAs were *not* included in the documentation sent to nurses.

### 2.2. Interviews

HCAs attended semi-structured interviews. Dimensions of the work environment identified by the UK Health and Safety Executive [31] were used as guidance in generating the interview schedule: *Demands* including workload and shift patterns; *Control* including the decision-latitude of employees to do their work; *Managerial support* that refers to encouragement and to job resources provided by the health Trust; *Peer support* that refers to encouragement and practical support from work colleagues (i.e., HCAs and nurses); *Relationships* that includes team working and inter-staff behaviours; *Role* with regard to employees’ understanding of their role and any conflict in that role; and *Change* with reference to the management and communication of organisational or practice change.

The HSE guidance provides the means of a generic evaluation of job demands and resources and so does not refer to specific aspects of any given job. For health care employees the emotional demands of caring are further sources of work demands [26]. When interviewed, HCAs therefore were initially asked to score on a scale of 1–10 their level of perceived stress when caring for patients, and what demands they faced in their day-to-day roles when working with patients.

Interviews with HCAs lasted approximately 45–50 min and were conducted at a mutually convenient time in a quiet room within the hospital but away from the wards. Consent was reaffirmed and, with permission, the interview was audio-recorded and transcribed.

Team work underpins care delivery primarily involving nurses and HCAs. As noted earlier HCAs expressed strong, negative views of nurses with regard to team relationships and so a post hoc decision was made to interview 10 nurses from the Unit to give better insights into team work issues. These were conducted in a mutually agreed location by author CU, audio-recorded (with permission), and transcribed. However, being focused only on interpersonal or inter-group issues, the interviews lasted around 25 min.

#### Focus Group (HCAs Only)

A focus group was convened to revisit the outcomes identified from individual interviews with HCAs, and to also provide opportunity for further insights to emerge in particular with regard to coping strategies. The group met outside of work hours in a room at the authors’ university. Consent was reaffirmed, ‘group rules’ established to ensure equity in responding, and attendees were reminded of the need for confidentiality. The meeting lasted for 50 min conducted by author C.C.U.; author A.M. acted as ‘scribe’ and did not take part unless directly asked a question.

### 2.3. Ethical Considerations

The main ethical issues were consent, confidentiality and anonymity in any dissemination of the study. Participants received a detailed participant information sheet and were also assured of confidentiality and of anonymity in any dissemination medium. Identities of responders were held confidentially in a locked store to which only author C.C.U. had access. For collation, all interview transcriptions of interviews were ascribed an identifier code. HCAs who responded to written invitation to attend a follow-up focus group were sent a new participant information sheet and consent form, and their details were securely held. Individuals were not identifiable in the transcripts.

Informed consent was obtained from all subjects involved in the study. Ethics and R&D approvals were obtained from the Faculty Research Ethics Panel (Faculty-DREP-17-003) and permission was granted by Research & Development Office of the Trust. Ethics approval was extended at a later stage to include participation by nurses.

### 2.4. Analytical Strategy

Audio-taped interviews were first transcribed verbatim. In view of the application of HSE guidance on work environments, and the job-demands-job resources model, the analysis of transcriptions adopted a framework approach to substantiate the main themes, with thematic analysis for secondary identification of sub-themes by following guidance from Braun and Clarke [32]. That process began with familiarization of data by carefully listening to the audio recordings and by using study notes made during data collection. Data were then coded by author C.C.U. and referred to discussion with co-authors for corroboration. Relevant quotes, information or descriptions were extracted and collated as sub-themes that emerged from the transcripts, and then located within the main themes (See Table 1).

#### Rigour

A quiet location for individual interviews was agreed that would enable interviewees to leave the ward for the required period. The focus group was conducted post-shift on university premises, again undisturbed. Analyses of all individual and group interviews applied established protocol and followed criteria for credibility and dependability [33] by at least two authors checking transcriptions, coding and collation of narrative extracts. Inclusion of a focus group of HCAs enabled corroboration of themes from the interviews, and confidence in data saturation.

## 3. Findings

Findings are reported in two sections. Section 3.1 presents outcomes of individual interviews with HCAs and of the subsequent focus group. Findings are collated according to Job demands and Job resources [21], with additional data on coping strategies (i.e., ‘Personal resources’ in the JD-R theory [25]) from the focus group. Nine themes are identified with identification of emergent sub-themes (underlined). Word length constraints meant that supplementary, supportive evidence, particularly from the focus group, have been included in Table 1.

Section 3.2 presents the outcomes of post hoc interviews with nurses conducted to illuminate further HCAs’ perceptions of teamwork in the Unit. Illustrative comments can be found in Table 2.

### 3.1. Individual and Group Interviews with HCAs

This section provides examples of comments and narrative that illustrate the themes.

#### 3.1.1. Job Demands


**Theme 1. Demands of caring**


Dementia care is highly demanding (see Background). HCAs were first asked to comment, on a scale of 1–10, their level of stress from working a shift on the Unit. There were times when the impact was ‘moderate’ but experiences were often very stressful:

*“…to be honest, sometimes it’s like 5 but sometimes it’s 10….they change like, every minute”*.(HCA 7)

*“Yes, I would say eh working in a dementia unit is a stressful job to do, it’s very stressful, yea… will rate it 9.5”*.(HCA 1)

Patient frailty and vulnerability meant that HCAs had to apply effort continuously across a whole spectrum of daily needs, making care especially challenging:


*“…vulnerability is a thing you have to look into…because [patients] are advanced in age, they are frail, so you have to put in extra effort and extra care”*
(HCA 5)


*“…patients obviously…need our support in nearly everything: toileting, bathing, washing, feeding and a whole lot, and you still must do laundry for them obviously on daily basis. You see, it’s difficult. You can’t compare it with other wards”.*
(HCA 15)

Patients with advanced dementia also tend to be resistive of day-day care activities complicated further by their unpredictable moods that could include abuse or sudden aggression. For example, in this study,


*“…sometimes, they don’t know you are helping them, and they will start fighting you”*
(HCA 10)

HCAs were mindful of potential risks to themselves:


*“Sometimes, I’m very concerned because some of the residents or patients have challenging behaviours… when you are going to work and you know you have such huge number of people to deal with and different presentations, yea, you get worried”*
(HCA 8)

Despite the demands and physical risks, the consensus of participants was that such demands and risks were to be expected. HCAs were altruistic about the challenges they faced whilst caring for patients, illustrated by the following comment:


*“…they’ve lived their lives and at old age they really need to be taken good care of. But I enjoy doing it. Personally, that’s what I like to do”*
(HCA 3)

Rather, the main work-related problems for participants were demands arising from aspects of the work environment. The following comment emphasised the effect this can have:


*“…the nature of the [work] environment affects how I feel after some shifts; stressed, angry, tired you know.”*
(HCA 3)


**Theme 2. Demands of a high workload**


HCAs identified high workloads which they attributed to inadequate staffing. HCA4 summed up the problem when asked ‘What would help to make your job, your role even more effective and interesting?’ by simply stating “*more staff*”. Understaffing was suggested to be related to cost-cutting, for example


*“…there used to be proper staffing but now maybe they are trying to cut cost. The issue of staffing has not been critically looked at”.*
(HCA6)

Participants emphasised that this led to work intensification, for example,

*“The demands of the job in a situation… where you are short-staffed then definitely you are going to be stressed where in a situation whereby a workload of 5 people is being done by 3 then definitely you will be stressed”*.(HCA 6)

The following comment from a very experienced HCA indicated that the dementia Unit was not unusual in this respect:

*“… the recommendation [in mental health care] is that dementia clients require more staffing support than others. I’m of the opinion that NHS hospitals do not provide that “*.(HCA 13)

Staff shortage also was perceived as detrimental to allocation of staff to teams on a shift as the shortage meant excessive reliance on temporary staff. For example,


*“We rely on irregular staff so much…that can be stressful on substantive staff. Not because they are not good at their jobs but because they are not substantive, they don’t know the routines”.*
(HCA 5)


*“Sometimes when you see the rota and see people you might be working with… you are so worried about it.*
(HCA 5)

Relatedly, temporary staff may introduce a skill mix profile that becomes unbalanced to the effect that HCAs were distracted by having to provide supervision:

*“Staff who don’t know the ward, or our patients, are a bit of a pain [problem] really. You’ve got to show them literally everything and that is difficult when you’ve got stuff to do really”*.(HCA 14)

Supporting inexperienced agency staff also can potentially increase risk of harm to HCAs in the team:

*“Sometimes you must restrain patients who are aggressive so you might have staff who has not enough training or has not been given an experience? If…. someone just let go, that patient will just hit you ”*.(HCA 5)

It was suggested that the manager appointing inexperienced agency staff should be more aware of the skill mix consequences:

*“I think it’s more on the people doing the staff mix. For instance, [the nurse] setting up your ward should know if you are bringing on an agency or someone who is not regular on the ward.”*.(HCA 14)

That issue extended to perceptions of nurses in charge who also lacked experience:

*“Lack of experienced staffing is across the board …You can have experienced domestics [HCAs]…that could make all the difference to a team…whereas a charge nurse with maybe no experience….yea, its lack of staffing, lack of experience”*.(HCA 4)


**Theme 3: Demands of poor shift patterns**


Staff shortage also had implications for the scheduling of shifts, with a greater application of short shift rotations. Such patterns were a major source of tiredness and exhaustion:

*“The shift patterns are horrible. We rarely have enough rest before going back to work. It’s stressful”*.(HCA 9)

*“The shift pattern isn’t healthy at all. It’s physically and mentally draining”*.(HCA 8)

Unsurprisingly, the scheduling of short shifts had an impact on family life, for example:

*“…I do not have enough out of work hours after a shift; knowing I’ll be going back to work in few hours and with the same patients and probably staff who are not very helpful, I think we just need more time off the ward for ourselves, families ”*.(HCA14)

#### 3.1.2. Job Resources

‘Job resources’ should help staff to meet their work responsibilities and stimulate personal and/or professional growth. This study identified deficits that introduced serious challenges for HCAs, particularly with regard to team leadership, in-team relationships and interpersonal factors, so adding to the demands they felt.


**Theme 4. Team leadership**


As support workers HCAs have little decision-latitude and so look towards nurses to determine the team rota and work allocation for the shift. Good communication therefore was essential in that respect:

*“The easiest shifts I have are the ones where communication at the beginning of the shift has been clear and concise……[so] you know exactly where you are and what you are doing and who’s doing what”*.(HCA 11)

However, poor leadership by nurses was a strong sub-theme for interviewees, as it left them feeling disorganised and confused as to their allocated tasks within the team. Imprecise communication caused a lack of role clarity,, for example:


*“The problem with the stress we face in most areas is when roles are not defined… When roles are not defined, this is when you see people playing around looking for what to do. Some dedicated ones are stressed, and it helps more when people know what to do, when to do it and how to do it”*
(HCA 11)

This participant summarised the situation succinctly,


*If you have no leadership then, you have no team really…’*
(HCA 6)

The effects of poor role clarity were exacerbated by HCAs’ lack of autonomy, also identified as an issue by other studies (e.g., Schneider et al. 2010; Cheloni & Tucker, 2019), for example:


*“…if a client is supposed to be on a higher level of observation due to their behaviour, and to keep other clients safe…I can’t put the client on a level of observation, like three to four, without a doctor or other professional group’s input”*
(HCA 6)


**Theme 5: Inter-relationships within the team**


HCAs were very vocal on issues of team-relations; judging by the number of comments and strength of views regarding this theme possibly was the most significant resource issue for them. Teamwork is reliant upon good team spirit and effective inter-relationships between team members:

*“Good team spirit helps a shift go well. Even if you are short staffed but the staff on ground are willing to work, it makes it go well”*.(HCA 4)

However, some HCAs identified poor team spirit in the team. Most commented very strongly that group-relations between team members were poor. This could be between HCAs themselves:


*“To be honest, some Healthcare Assistants shy away from work. They are lazy and do unimportant stuff, leaving the important ones…Even the nurses themselves don’t like working with them, and this is common among permanent staff”*
(HCA 13)

However, the general opinion of HCAs was that the most significant relationship difficulties were with nurses. One prominent complaint was that nurses were often absent from the ward so unsupportive of HCA colleagues. Time spent in the office means less time on the ward, which some HCAs blamed on authorities who demand a high amount of paperwork from the nurses they work with. For example,

*“Nurses are under pressure because after shifts some tend to stay and ensure that every paperwork is completed. Apparently, this is pressure coming from top management, CQC [Care Quality Commission] and government”*.(HCA 12)

Most interviewees however were sceptical of this explanation and suggested that completing paperwork was an excuse for avoidance of ward work as it provided a convenient distraction for nurses to minimise their time on the ward. For example:

*“Also, there is …this mental disposition by some nurses that they are only there to do medication…after doing medication they…will go and probably sit down and be doing paperwork and they will leave the rest of the work to the support worker”*.(HCA 6)

One HCA claimed that the diversion of paperwork potentially could have detrimental outcomes, noting quite graphically:

*”Could you imagine nurses leaving patients in need just to update their paperwork. But this happens all the time. Not nice at all.”*.(HCA 2)

Further, HCAs also considered that this situation was aggravated by poor inter-personal skills of some nurses:

*“When you are struggling, you won’t want to ask for assistance because the outcome or the way the person will turn you down will make you feel inferior or incapable, so you won’t want to ask.”*.(HCA 6)


**Theme 6: Lack of Support (Nurses and Managers)**


Perceptions of poor support permeated HCAs’ views of the work environment, evident in some examples already given above: poor shift work patterns, poor leadership, lack of role clarity. A further strong issue for HCAs was that nurses made a clear demarcation as to their role:

*“Some…clearly dictate the role ‘I’m a nurse, I’m a nurse’. Even when you are short on the floor, they are not really ready to help and it’s really, really exhausting to see someone not working, not busy but still disturbing you whilst you are carrying out your own responsibility”*.(HCA 7)

This caused frustration and irritation as it was interpreted as nurses not engaging with personal care for patients:

*“Sometimes even when HCAs are short staffed… these nurses don’t help.”*.(HCA 5)


*When they are not very busy, what stops them from toileting a patient?*
(HCA 7)

Additionally, a perceived poor concern from managers for staff members’ personal well-being was strongly criticised by some. As an example, one HCA was highly critical of a lack of concern or acknowledgement of effects on him following an injury sustained in caring for the patient:

*“I had no support, nothing No. I was left literally…seeking support somewhere else from other than my own team”*.(HCA 4)

This lack of concern was further illustrated by what HCAs considered an abuse of the Bradford ‘Score’, a formula commonly used as a human resource tool to measure the extent of an individual’s absenteeism. Participants perceived a deliberate misinterpretation of the ‘Bradford score’ by managers who they claimed used it as a threat to staff, so causing some to come to work even when they are truly unwell:

*“I’ve seen people coming to work…that they are unwell, that they shouldn’t have come in, but they are worried about their Bradford Score. It’s like a threat, it’s awful. I can understand why it’s in place and I can understand the reasons behind it, but it’s dealt with very badly”*.(HCA 5)


**Theme 7: Change**


‘Change’ is a source of stress for employees who are not consulted about impending significant changes to their workplace, and subsequently their roles (Cousins et al. 2004). Participants appeared reconciled to not being consulted about change in practices that may affect them. One focus group member suggested a feedback forum might be useful

*“where one can put in your complaint, one can put in your observation… not only a suggestion box… but a functional one”*.(FG Participant)

However, others were sceptical and noted that even making a complaint or suggestion was unlikely to be taken on board by managers:

*“If we say something [then] nothing would be done and if you keep on saying, your name would be crossed off if you are bank and not substantive staff. We’ve said things and nothing was done”*.(HCA 12)

#### 3.1.3. Coping (Personal Resources)

In addition to corroborating individual comments, focus group participants were asked to comment on how they collectively manage their stress in what at times was clearly a tense environment. HCAs appeared reticent to discuss coping strategies suggestive of a lack of availability of anything constructive. Indeed, comments pointed to individuals finding their own way largely through passive approaches. Two themes emerged: Acceptance and Work-home balance.


**Theme 8: Acceptance**


HCAs appeared reconciled to the reality of working on the Unit. The main coping strategy they adopted was a pragmatic acceptance of challenges within a difficult job. The following comment summed up the discussion:

*“You just get on with it, you got to get on with it. You can’t run away, can you? You don’t have any choice; the work must be done. You just get on with it”*.(FG participant)

Staff sought frequent, short breaks from the ward as time-out helped them to cope, albeit providing only brief moments of relief. For example,

*“I take 5-minute breaks at work and that helps. I think more breaks should be encouraged in this job really, it’s so stressful”*.(FG participant)


**Theme 9: Work-home balance**


Others felt that they were able to compartmentalise work and home life:

*“I don’t take it home. It’s not that you don’t care, obviously you have done what you could as soon as I leave I just so yeah someone else is taking them”*.(FG participant)

However, HCAs were not forthcoming as to just how, or if, they were able to mentally ‘switch-off’ as claimed. Two focus group participants suggested a more constructive approach by trying to anticipate problems and take steps to ensure that unresolved care issues were not left behind after a shift (see also Table 1).

*“… if I know I have the same shift the following day and I’m having the same group of people that I know during the previous shift. I orientate… them to what they should do, I’m comfortable and I’ll …go home and relax”.*.(FG participant)

For two group members, though, respite was acknowledged as being only temporary and they sought a means to help reduce the impact of returning to the workplace:

*“When I leave, I leave and then as soon as I walk through that door, the stress returns. So, if it’s unresolved, then I bring it back to work either by talking it through with colleagues or going to my management team”*.(FG participant)

### 3.2. Post Hoc Interviews with Nurses

Comments from HCAs with regard to team working and inter-staff relations reflected poorly on the nurses in the Unit. Accordingly, this study adopted recommendations [29,30] that both staff groups be included. Interviews with 10 nurses from the Unit therefore were introduced to clarify those issues.

Nurses were first asked contextual questions in order to identify if their general viewpoints on patient care related to those expressed by HCAs. As with HCAs, the 2009–2018 period also evidenced a decrease in nurses; around the time this study was undertaken there were 35,000–40,000 NHS nurse/midwife vacancies, equivalent to around 11% of posts [11,34]. Understaffing was identified as a significant factor on nurses’ workload:


*“…ehm overall, I think it’s more of nurse staffing on the ward that’s causing a whole lot of problems”*
(Nurse 4)

Nurses also cited patient unpredictability. For example,

*“The worst part of it is that sometimes, they don’t know you are helping them, and they will start fighting you”*.(Nurse 10)

Nurses also were asked to comment on the level of stress on a scale of 1–10 and all agreed that they found the workplace stressful. For example,

*“Yes, it’s stressful. I’ll put between 9 and 10”*.(Nurse 7)

Whilst many comparisons cannot be made between staff who have different roles and responsibilities, such comments suggest that levels of tension and work pressure had a degree of commonality in these respects.

The intention of these interviews was to seek better insights into the team working difficulties identified by HCAs from their perceptions. Nurses at times were complimentary of their HCA colleagues but like many HCAs mostly they were negative as to inter-staff relationships, though their perceptions of the sources of such discord were very different (see Table 2). It would appear that misunderstandings of the others’ roles and responsibilities were widespread, suggesting that there were communication and attitudinal issues from both groups that impacted on the team working ethos.

## 4. Discussion

This study evaluated experiences of health care assistants (HCAs) working in an inpatient dementia Unit of a UK National Health Service (NHS) hospital after around 10 years of funding cuts and staff reductions through financial austerity measures in the UK (2008–2019) [10]. Difficulties with staff recruitment and attrition in the UK were already recognised prior to the austerity period [15] and they considerably worsened after 2008. The objective of this study therefore was to identify work environment factors for HCAs at the end of 2018 and to relate them to the Job Demands-Resources theory [21] of work stress. According to this theory, if demands and resources are in balance then the adequacy of job resources may act to ‘buffer’ the impact of demands on staff. Present findings identified staff shortage and high demands in the Unit but also significant resource deficits indicative of a very stressful environment and risk of staff ill-health, in particular burnout.

Delivering in-patient dementia care is demanding and stressful [2,3,17], and evidenced in this study. Despite those challenges HCAs were altruistic about the demands of their caring work, consistent with findings of others [14,20], though taking frequent breaks of 5 min or so was an important aid to coping, also noted by Bailey et al. [17]. HCAs were mostly critical of the demands arising from their work environment, many of which were associated directly or indirectly to understaffing and subsequent high workload, exacerbated by skill-mix issues from an excessive reliance on inexperienced agency staff appointed to fill the staffing gaps, and by detrimental effects of understaffing on shift schedules. The situation was worsened by significant deficiencies of job resources in particular a poor collaborative ethos as a consequence of perceived poor team leadership by nurses in charge, poor team dynamics, and lack of mutual respect from nurses. Early in the period of financial austerity Schneider et al. [14] had found that whilst there were ensuing concerns over staffing levels, unpredictable patient behaviours and lack of recognition of HCAs experience, the teamwork ethos and collaboration was strong and enabled efficient care delivery. Around the end of the austerity period, however, the study by Cheloni and Tinker [20] reported that HCAs had become demotivated by organisational and environmental factors, including staffing levels, suggesting workplace issues may have worsened in the interim period. The current findings of serious discord between HCAs and nurses working together in a team supports that position.

If resources are inadequate then they can themselves become additional demands and hence sources of further stress [21]. In that respect it is notable that HCAs were especially critical of interpersonal skills of some colleagues and inter-group relationships especially with nurses. Good leadership from nurses, and engagement of HCAs with work that is allocated to them, can work well for inter-group relationships [14] but in this study a teamwork ethos was perceived by HCAs regarding their nurse colleagues. On reflection, such highly critical views of the HCAs made it important to also consider the nurses’ viewpoint. Nurses were largely defensive and emphasised the importance of protecting the requirements of their professional registration, which in their view, was not acknowledged by HCAs. In their role they identified a responsibility to complete excessive administration and, because of shortage of staff, identified they had to spend frustratingly long periods of time supporting patients’ family members. Such activities took them away from the ward and could give an impression of being unsupportive of colleagues, but many HCAs interpreted nurses’ absences as being avoidance. HCAs also felt that leadership and communication by nurses was poor, and that HCAs’ practice experience was not respected by them. Collectively, such views were not recognised by nurses, suggesting a dissonance and deep-seated disintegration of mutual understanding and trust which was causing inter-group discord and a lack of mutual support and respect between the two staff groups. Poor relationships amongst care teams may foster intimidation, impoliteness and complaints [35] and this very much appeared to be evidenced in the hospital Unit in this study.

One frustrating factor for HCAs, similarly documented by Schneider et al. [14] and Lloyd et al. [19], was that their experience of practice was not acknowledged by nurses or managers, making them feel marginalised. HCAs are key workers in providing direct patient care and in this study they identified frustrations they felt at not having decision-latitude which. if enabled, would allow HCAs to make a contribution to work planning. A proposal by Schneider et al. [14] that training of experienced HCAs would enable them to engage more with clinical issues for individual patients had not been adopted in this dementia Unit. It would seem therefore that little has changed in the interim, a position also noted by Cheloni & Tinker [20].

The Job Demands-Resources model [21] suggests that there is always risk of high stress in workplaces where there is an imbalance between high job demands and low job resources. Current findings point to HCAs being exposed to considerable demands but with serious deficits of job resources required to buffer the effects on HCAs’ well-being. In a follow-up analysis of data from the study by Schneider et al. [14], those authors (Lloyd et al. [19]) reported observations of, and interviews with, health care assistants in dementia care and identified them as having a strong social identity with a strong commitment to their group. Their solidarity fostered bonding of the group but this also emphasised their feelings of marginalisation, but also limited their communication with other staff on the wards so exacerbating a feeling of exclusion. These elements are all evident in the present study, from health care assistants and also the nurses they work with, but relationships appear to be more strained, confrontational and divisive. Such imbalance predisposes workers to ill-health as a consequence of difficulties in completing their work [34], potentially leading to emotional disengagement, self-isolation, decreased productivity and eventually burnout [36,37]. If not dealt with, workplace issues can accrue over a period of time and many aspects in this study appear to have done so. The extent of imbalance in the JD-R evaluation identified by this study in the dementia care Unit warrants urgent intervention.

### 4.1. Strengths and Limitations

This study provided a comprehensive evaluation of work environment and stress experiences of HCAs towards the end of national financial austerity measures in the UK (2008–2019). The sample size was modest, though not unusually so for a qualitative study. Additionally, high tensions on the wards may have led some participants to self-select on the basis of negative experiences. This could not be ascertained but the interview sample represented 42% of the estimated number of HCAs either employed substantively or as frequent agency appointments, so likely providing a reasonable profile of the Unit. Further, the follow-up focus group of HCAs incorporated participants who had not been interviewed so giving some confidence in the key themes emerging from the interviews.

Health care delivery is sensitive to socio-political issues and to local context [26] so focusing on this single study site means that findings cannot be generalised. However, the study site was not atypical with regard to difficulties in recruiting and retaining HCAs. Impacts of staff shortage were evident in this study and some issues it identified seem likely to have a degree of resonance with situations elsewhere.

Additionally, post hoc introduction of interviews with 10 nurses gave added-value as it helped to contextualise team working issues highlighted by HCAs. Though not part of the original design their interviews provided important additional insights into the inter-group relationships between HCAs and nurses.

### 4.2. Implications and Relevance to Practice

Current findings seem likely to have resonance with other dementia hospitals also seeking to reverse the effects of high staff attrition/inadequate recruitment that has been observed across the NHS in recent years (pre-COVID pandemic). Staffing, team working and leadership are critical to collaboration and quality of care but were deficient and appeared to exacerbate the recognised challenges inherent in delivering dementia care. Review [26] suggests that the period of 2007–2013 saw some stressors emerge as priorities for health care staff, including scheduling of work shifts and deteriorating interpersonal relationships that appear as common issues of both job stress and job (di)satisfaction. The author suggests that work–life interference and social relations may be sensitive to the changing workplace environment, particularly understaffing, so perhaps causing a worsening situation for HCAs since 2010 when Schneider et al. reported positive team working at that time. The very challenging social relations identified in the present study support this.

Staff shortages are unlikely to be resolved in the near future, especially following impacts of the COVID pandemic. In the interim, interventions could focus on improving psychosocial job resources for HCAs and nurses. A message from this study is that managers should promptly address emerging problems and so defuse subsequent tensions that are counter to the functioning of dementia care teams. Further, consideration ought to be given to the experience of HCAs by engaging them more in planning and delivering their work, and to better support for their personal stress management

## 5. Conclusions

Caring for patients with dementia introduces considerable emotional demands for staff but it was the work environment that was more of an issue for Health Care Assistants (HCAs). In this study, work environment demands appeared related to understaffing leading to high workloads, poor shift patterns and skill mix issues caused by excessive reliance on temporary, agency staff, some of whom had not the experience of substantive employees. Deficits in job resources for HCAs were perceived to be poor team leadership, poor inter-group relationships, lack of role clarity, and lack of support especially from nurses in charge of the team. Additionally, HCAs’ status as unqualified support workers was frustrating and, in their view, experienced HCAs could contribute to case discussions as appropriate.

According to the Job Demands-Resources model the significant imbalance caused by high job demands and serious deficits in job resources translated into a tense and very challenging workplace that risked stress-related ill-health for HCAs. Taking a dual approach by applying assistive interventions to improve personal resources of health care assistants but at the same time improve their job resources by promoting the psychosocial environment, especially of inter-personal and professional relationships, enhancement of social support and management support, and decision-latitude. These are more immediate, amenable interventions. Present findings from this local application of JD-R theory to an understaffed specialist dementia care unit reinforces such strategy as a useful first step to improving collaboration and team working between health care assistants and nurses.

## Figures and Tables

**Table 1 ijerph-20-00065-t001:** Supplementary quotes from HCAs illustrative of examples of sub-themes arising from individual and focus group (FG) interviews. See also the text.

Theme	Sub-Theme	Examples	Comment
**Job Demands**			
Demands of Caring	Patient frailty and vunerability	*“…you’re trying to understand…and the patient is in a different world”.* (FG participant)	Patient confusion as a challenge to HCAs.
	Unpredictable patient behaviours	“*With dementia clients or patients…they have varying characters depending on what might trigger each action at any time. …. they are unpredictable*” (HCA 6). “*…I always have it in mind that I should prepare for the unexpected, unlike when I go to other wards that are not dementia-related*” (HCA 3). *“… what happens is* [those with dementia] *have challenging behaviours so when you want them to sit, they want to be up, when you want them to go to bed, they want to stay awake all night”.* (FG participant)	Unpredictable triggers of adverse behavioural change.
Workload	Understaffing	*“It’s not very nice to have a ward of say 20 to 30 patients, for instance, and you have two HCAs to get them up.” (*FG participant)	A demanding HCA:patient ratio. Workload intensification.
	Skill mix profile	*“If you have inexperienced workers…you end up teaching them other than having them assisting with the work”* (FG participant)	Distraction and time-consuming supervision of temporary staff.
Poor Shift Working Patterns	Tiredness/exhaustion	*Shift patterns should be like weeks of earlies, or weeks of lates, and not early, late, early again… we don’t get enough rest in-between shifts*” (HCA 9). “*… shift patterns are awful. I usually don’t sleep well before going back to work in the morning after a late shift because you keep thinking you’ve got work in few hours ”* (FG participant) *“It would be easier for us to do long days than half days daily. It’s killing. It’s so difficult.”* (FG participant).	Poor recovery post-shift. Risk of ill-health.
**Job Resources**			
Team Leadership and Management	Poor team management	*“ If they* [the nurse handing-over] *discover that the service user’s presentations need more help then…. he’s supposed to tell probably the Bleep Holder, or the ward manager, so that they provide extra staff. But in a situation whereby* [the] *nurse …did not make adequate provision then*…[those] *being handed over to will be short staffed.”* (HCA 6).	Unmet need for adequate staffing of a team.
	Poor role clarity	*“.it comes to the shift co-ordinator…the moment you step in after the taking and handing over, they say ‘okay this is what you should start, you have a goal for that day, this is what we must achieve’. But you have others who* [encourage] *free will: ’… everyone just do something, make sure everyone is safe’. When you have such people the roles are not defined”.* (FG participant)	Imprecise instruction and guidance from nurses in charge.
Inter-Relationships within the Team	Group relations	*“*It [team work] *depends on who you work with and obviously the situation and the environment…”* (FG participant) “*I always feel anxious before going to work when working with* *staff who aren’t good team players*” (HCA 12). “*Once they* [HCAs] *become APs* [Associate practitioners, a stratum of support staff located between HCAs and nurses] *, they don’t want to do any personal care, they just want to do the meds and ehm, they feel like they are staff nurse”,* (FG participant)	Recognition of the situation and impact on individuals before work. Discord between HCAs and related staff groups.
	Avoidance	*Some senior staff members sit in the office all day doing paperwork. And, when the staff are sitting in the office doing the paperwork, they’ve got no idea what is going on. Because they are in their own little bubble”.* (FG participant) “*I think when people aren’t working as a team you are left to do everything by yourself whilst other people are sitting in offices. Especially in the mornings when you have got to get people up washed and dressed, you always end up getting same ones sitting in the office and the other people doing all the work”.* (FG participant)	Pressure on nurses to complete voluminous paperwork. Poor timing when HCAs likely to need support.
	Inter-personal skills	*“The way nurses delegate roles matter a lot to us in this job. Some don’t have good interpersonal skills. They delegate role to you as if you are a nobody, as if you are robot.*” (FG participant)	Perceived attitudinal issues.
Lack of Support (Nurses and Managers)	Demarcation as to role	*“Most qualified nurses…feel the work is only for the HCAs*”. (FG participant) *What stops them* [nurses] *from feeding a patient? …why must they always delegate personal care to HCAs? ”* (HCA12).	Suggestive of hierarchical relationships regarding tasks.
	Lack of support from colleagues and senior staff	*“I had no support, nothing No. I was left literally…seeking support somewhere else from other than my own team”* (HCA 4)	Lack of acknowledgement of injury from a patient, and its impact.
**Coping**			
Acceptance		*“You just get on with it and* [on night shift] *pray that daybreak comes quickly”.* (FG participant)	Reconciled to the reality of the ward
Work-home Balance	Compartmentalise work and home life	*“, I never take anything home. As soon as I walk out that door, I don’t need to think about patients or what’s going on, no”.* (FG participant) *“I certainly don’t sit at home thinking about the patient”.* (FG participant)	Distinction of responsibilities at work and at home
	Anticipate problems	*“…when I’m seeing another team…who might be experiencing the same thing as I did that night then I can also talk to those taking over from us to have a review… if there is something they can do, maybe an extra staff or swap with another regular. In that way when I’m coming back, I know the work is going to be easier than it was in the previous shift… I did that a few times”.* (FG participant). “ *On my off days, I take my mind off work, for me to relax. Coming back to work, I prepare myself ahead of the shift in case the staffing strength is low. I prepare myself psychologically to go through my shift*” (FG participant).	Anticipation of the next shift

**Table 2 ijerph-20-00065-t002:** Comments from post hoc interviews with nurses.

Theme and Sub-Themes from Interviews with HCAs	Perceptions from Nurses	Comment
**Theme: Team Leadership and Control**		
Poor team management/poor leadership	*“Sometimes you have these HCAs…they come down, the shift goes smoothly, they clean the board, get the book, bring the patients out, do their laundry…some. Don’t even wait for you to direct them, they go straight* (and) *before you know it, they’ve finished*” (Nurse 6) *“The only people we have problems with are the HCAs because some of them don’t want you to tell them things to do”* (Nurse 1)	Leadership can be effective depending upon application by the HCA HCAs can be unresponsive to work allocation from nurses
Lack of role clarity	*You as a nurse, you can delegate duties, but HCAs won’t do them.*” (Nurse 10)	HCAs can be obstructive
**Theme: Inter-relationships within the Team**		
Team spirit	*“…when you delegate work to HCAs, some of them don’t do it whilst some will do it reluctantly. It’s just so difficult.”* (Nurse 9) *“If you don’t have good personal relationship with HCAs they can frustrate you …some have a lot of ego and always want to show you they know; no team support.*” (Nurse 10)	HCAs not supporting a team ethos
Inter-relations	*“When you interact with some of these HCAs…about 80%… think the nurses disrespect them”* (Nurse 8)	Misconception from HCAs
Poor inter-personal skills	*“…you are here to protect your registration, you are there to do your job, so you don’t need to be told what to do by an HCA.* (Nurse 7). *“Some HCAs are even ‘bosses’…they’ve been there for 30 years. They will tell you all…the charge nurses that they’ve worked with, so where are you coming from?…Then you have to tell them ‘hey boss, tell me what you want to do this morning?’ You have to just say it that way otherwise they won’t do anything”* (Nurse 5).	Negative interactions and communication between nurse and HCA
Demarcation as to role	*“They* [HCAs] *could do anything and get away with it but I’ve got my PIN* (registration) *to protect. Some of them do understand and appreciate that. They can understand why you do certain things. But some of them, most of them, I can tell you, they think that they are not being respected”* (Nurse 8). *“With the HCA, it’s somebody who’s been there for 15 years, and you’ve just done University for 3 years, now you are commanding them”* (Nurse 6) *“Few understand the fact that you are responsible”* (Nurse 10)	Professional responsibilities of nurses often not acknowledged by HCAs
**Theme: Lack of Support (from Nurses)**		
Absence from ward	*“…we cannot manage on the ward. We are so short of nurses…”* *“it’s just so difficult to manage, 17 to 18 patients on the ward and you still have families to deal with. Sometimes, we spend a whole hour or more attending to family needs for the people…without helping people on the floor to do other things”* (Nurse 3) *“…some nurses are so lazy, claim to be doing paper work that never finishes and not helping other staff on the floor, yea… its really that bad”* (Nurse 3). *“…these patients bump into each other and they fight…and you still have staff on the ward reading Newspapers. …If you have a ward like that people are not observing… There’s no way you should…take your eyes off these patients.”* (Nurse 10)	Impact of understaffing, responsibility to support families. Corroboration of HCAs’ claims of avoidance by some nurses

## Data Availability

The data presented in this study are available in Table 1.

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
