# Peer review of "Utilization of Job Demands-Resources (JD-R) Theory to Evaluate Workplace Stress Experienced by Health Care Assistants in a UK In-Patient Dementia Unit after 10 Years of National Financial Austerity (2008–2018)"

_ijerph, 2022, doi:10.3390/ijerph20010065_

Round 1

Reviewer 1 Report

Dear authors,

Thank you for the opportunity to get acquainted with the results of your research.

The introduction contains detailed information about the relevance of the study in terms of the reason for studying this particular professional group, at the same time it is necessary to give an overview of the approach used in this study - Job Demands-Resources theory. Conduct an analysis of studies where this approach has been implemented in relation to medical workers. To demonstrate its importance for the purpose of this study.

Sample. The authors describe in detail the sampling procedure, at the same time, the sample size is very small - 15 participants, even under the condition of a qualitative research paradigm. The authors are invited to either expand the sample - for example, through comparison with the results of medical personnel working in other departments (comparison groups). Or the authors need to justify the sufficiency of this sample size (demonstrate the criteria used for its calculation) with references to the relevant documents and articles. Also, the authors did not present the demographic characteristics of the sample (age, experience, education, etc.).

Despite the fact that the authors prescribe the procedure for collecting and analyzing the material, it is not entirely clear what methods of qualitative data analysis were used?

In the results of the study, a qualitative description of the identified topics is made. At the same time, the article lacks any illustrations (graphs and tables), which makes it difficult to see the results of the study in a comprehensive manner.

There are also shortcomings in the design of the article and the list of references.

In connection with the above, the article can be recommended for publication after significant revision.

Best regards, reviewer

Author Response

  1. Elaborate on the JD-R theory. This has been addressed - see Line 114-on
  2. Provide more data; expand to other units. The purpose of the study was to explore experiences of support workers specifically in a specialist dementia unit. This would have entailed recruitment from another hospital where conditions are likely to be similar but not exactly the same. There was no intention for it to be a comparative study rather to focus on a single unit. 
  3. Justify the sample size. This has been addressed, and referenced. See Line 155
  4. There are no demographic details. In retrospect  this might have been included for completeness. However it is not unusual in a relatively moderate, exploratory, qualitative study of this type (e.g. see Table 1 in Lloyd et al 2011 referenced in the paper; a study funded by the NIHR). The sample represented around half of all HCAs and nurses registered with the unit and there was no obvious divergence from the unit staff profiles. Gender and age were not were not raised as issues in the interviews.
  5. What methods of data analysis were used. The process is outlined in the Analytical strategy section. As noted the process followed the Braun & Clarke guidance for thematic analysis. Although a framework approach was utilised to link to Health and Safety Executive guidance for workplace evaluations, the dimensions were applied to help collate and format the Findings, Sub-themes emerged from the analysis and the framework revised slightly as a consequence. Table 1 gives some examples of this.
  6. The article would benfit from inclusion of tables. I think I may have made an error in using the submission template as the paper does include, and refers to, two tables. I have copied these into the comments in the manuscript but in case this has challenged the formatting I have also copied them in Worf into an email to the Editor.

Reviewer 2 Report

Dear Authors,

Congratulations for dealing with so socially important issue. I would suggest to change the title as the study group was small. In my opinion description of recruitment and ethical consideration might be shortened. In results section appropriate  analysis of data should be performed instead of detailed description. In such a small group results shouldn't be presented as a percentage.

Author Response

  1. Change the title as the groups are small. This has been done. Please see Lines 4 and 5 of the manuscript.
  2. The description of recruitment and ethics could be shortened. We agree - the presented narrative was along the lines of guidance in some of the tools now available to evaluate quality of reporting. This can lead to manuscripts being somewhat wordy. The revised text in Lines 208-on is shorter but still identifies the main points of consent, confidentiality, and anonymity. 
  3. Analysis of data should be presented instead of detailed description.The downside of applying a framework approach as the first step to analysing the data is that the main categories/themes are pre-set by guidance, in this instance from the Health & Safety Executive and the Job Demands-Resources model. However, the thematic analysis we applied to the transcripts also allowed sub-themes to emerge that when collated identified the impact that understaffing has had on the Unit, especially relationships between the HCAs and nurses. Table 1 gives some illustration of this as does the Discussion.

Round 2

Reviewer 1 Report

Dear authors, thank you for making corrections and finalizing the manuscript. Key remarks have been eliminated by you.

Best regards, reviewer